# In Silico Study Identified Methotrexate Analog as Potential Inhibitor of Drug Resistant Human Dihydrofolate Reductase for Cancer Therapeutics

**DOI:** 10.3390/molecules25153510

**Published:** 2020-07-31

**Authors:** Rabia Mukhtar Rana, Shailima Rampogu, Noman Bin Abid, Amir Zeb, Shraddha Parate, Gihwan Lee, Sanghwa Yoon, Yumi Kim, Donghwan Kim, Keun Woo Lee

**Affiliations:** 1Division of Life Sciences, Division of Applied Life Science (BK21 Plus), Research Institute of Natural Science (RINS), Gyeongsang National University (GNU), 501 Jinju-daero, Jinju 52828, Korea; rabia.mukhtar.rana@gmail.com (R.M.R.); shailima.rampogu@gmail.com (S.R.); zebamir85@gmail.com (A.Z.); parateshraddha@gmail.com (S.P.); pika890131@gmail.com (G.L.); jsyoon0517@gmail.com (S.Y.); yumikim@gnu.ac.kr (Y.K.); donghwanz@naver.com (D.K.); 2Division of Life Science and Applied Life Science (BK 21), College of Natural Sciences, Gyeongsang National University, Jinju 52828, Korea; noman_abid@gnu.ac.kr

**Keywords:** methotrexate, drug resistance, human dihydrofolate reductase, pharmacophore modeling, virtual screening, molecular docking, molecular dynamics simulation.

## Abstract

Drug resistance is a core issue in cancer chemotherapy. A known folate antagonist, methotrexate (MTX) inhibits human dihydrofolate reductase (hDHFR), the enzyme responsible for the catalysis of 7,8-dihydrofolate reduction to 5,6,7,8-tetrahydrofolate, in biosynthesis and cell proliferation. Structural change in the DHFR enzyme is a significant cause of resistance and the subsequent loss of MTX. In the current study, wild type hDHFR and double mutant (engineered variant) F31R/Q35E (PDB ID: 3EIG) were subject to computational study. Structure-based pharmacophore modeling was carried out for wild type (WT) and mutant (MT) (variant F31R/Q35E) hDHFR structures by generating ten models for each. Two pharmacophore models, WT-pharma and MT-pharma, were selected for further computations, and showed excellent ROC curve quality. Additionally, the selected pharmacophore models were validated by the Guner-Henry decoy test method, which yielded high goodness of fit for WT-hDHFR and MT-hDHFR. Using a SMILES string of MTX in ZINC^15^ with the selections of ‘clean’, in vitro and in vivo options, 32 MTX-analogs were obtained. Eight analogs were filtered out due to their drug-like properties by applying absorption, distribution, metabolism, excretion, and toxicity (ADMET) assessment tests and Lipinski’s Rule of five. WT-pharma and MT-pharma were further employed as a 3D query in virtual screening with drug-like MTX analogs. Subsequently, seven screening hits along with a reference compound (MTX) were subjected to molecular docking in the active site of WT- and MT-hDHFR. Through a clustering analysis and examination of protein-ligand interactions, one compound was found with a ChemPLP fitness score greater than that of MTX (reference compound). Finally, a simulation of molecular dynamics (MD) identified an MTX analog which exhibited strong affinity for WT- and MT-hDHFR, with stable RMSD, hydrogen bonds (H-bonds) in the binding site and the lowest MM/PBSA binding free energy. In conclusion, we report on an MTX analog which is capable of inhibiting hDHFR in wild type form, as well as in cases where the enzyme acquires resistance to drugs during chemotherapy treatment.

## 1. Introduction

A major complication in cancer treatment with chemotherapy is the development of resistance to previously effective drugs. Clinically, two main types of drug resistance exist: intrinsic resistance, which is not associated with drug exposure, but rather, with an innate ability of tumor cells; and acquired resistance, which occurs after exposure to the drug [1]. Various mechanisms like increased rates of drug efflux, alterations in drug metabolism, variations in drug targets, increased target expression, activation of survival pathways, increased expression of anti-apoptotic proteins and mutation of drug targets are involved in acquiring resistance to chemotherapeutic agents [2].

Human dihydrofolate reductase (hDHFR) catalyzes the reduction of 7,8-dihydrofolate (DHF) to 5,6,7,8-tetrahydrofolate in a nicotinamide adenine dinucleotide phosphate (NADPH) dependent manner. Tetrahydrofolate is an essential cofactor in several metabolic pathways like purine and thymidylate biosynthesis, playing a vital role in cell division and proliferation [3]. Due to the significance of its crucial role in nucleoside biosynthesis, hDHFR has widely been studied and exploited as a drug target [4,5].

Methotrexate (MTX) (C_20_H_22_N_8_O_5_) is an antimetabolite, an analogue of folic acid and a derivative of aminopterin antiproliferative drugs that inhibits dihydrofolate reductase [6]. The drug primarily penetrates intracellular targets through an active carrier transport mechanism which is shared by reduced folates and facilitated by the reduced folate carrier (RFC) [7]. This process is carried out by the enzyme folylpolyglutamate synthetase (FPGS) through the accumulation of glutamate residues [8,9]. MTX and polyglutamylated conformations of MTX are tightly bound inhibitors of hDHFR and hinder pyrimidine, and hence thymidylate biosynthesis [10,11,12]. Decreased MTX affinity has been detected in cell lines exposed to increased dosage causing mutations in hDHFR [13,14,15,16,17,18]. Mutations in dihydrofolate reductase variants with amino acid substitutions at residues Phe31 [19], Arg70 [20], Leu22 [21,22], Val115 [23] and Phe34 [24] existing in folate binding site have been detected in MTX-resistant cancer cell lines.

MTX-resistant point mutant hDHFR crystal structures have provided an understanding of the details of decreased binding of MTX or other antifolates [25,26,27,28,29,30]. Volpato et al. reported a combinatorial MTX-resistant hDHFR variant F31R/Q35E which exhibited >650-fold decreased binding to MTX to reveal the structural details of MTX resistance in the F31R/Q35E variant, and obtained the crystal structure of this variant bound with MTX at 1.7 Å resolution (PDB ID: 3EIG) [31]. This highly MTX-resistant variant is an effective selectable marker for several mammalian cell types, along with murine hematopoietic stem cells [32]. Since mutations triggering MTX resistance have not been studied in mammals, and MTX is an approved drug for human treatment, engineered resistant DHFRs provide highly capable ex vivo or in vivo selective markers for human [33].

In recent decades, advances in computational techniques have led to an acceleration of drug discovery [34]. For example, cheminformatics allows us to understand and characterize the molecular properties and chemical activities of specific compounds and produce huge libraries of small molecules to screen against particular therapeutic processes [35]. After candidate identification, other cheminformatics techniques can be utilized to generate libraries of compounds which are structurally and chemically similar to the identified “hits” in order to improve stability, toxicity and kinetics. Additionally, bioinformatics methodologies can be applied to study the therapeutic activity of candidate drugs predicting interactions between drugs and proteins, to analyze the impact on biological pathways and functions and to determine genomic variants that may vary drug response [36]. Accordingly, several approaches have been developed to reduce the research expense and risk of failure for drug discovery, among which computer-aided drug design (CADD) is one of the most effective [37].

Since drug resistance is hinders chemotherapy, there is an urgent need to discover the drugs that inhibit hDHFR in wild type as well as in mutant form, i.e., after acquiring resistance to MTX. Singh et al. developed a small peptide as an anticancer drug targeting hDHFR which was supposed to be effective in MTX-resistant hDHFR because of a larger drug–protein interaction area [38]. Despite the larger interaction area of the peptide, it was specifically designed to inhibit only wild type hDHFR. We carried out a computational study to identify a candidate molecule capable of inhibiting wild type along with mutant hDHFR. Structure-based pharmacophore modeling was performed exploiting hDHFR wild type and drug-resistant F31R/Q35E variant structures in complex with methotrexate to allocate important chemical features of protein-ligand interactions. Pharmacophore models, WT-pharma and MT-Pharma, with four features comprising key residues were selected from ten models generated for each structure. The selected pharmacophore models exhibited the highest area under the receiver operating characteristics (ROC) curve, verifying the sensitivity of the models to retrieve active compounds. WT- and MT- pharma were further subjected to validation by the Guner-Henry decoy test method.

ZINC was initially developed as an open-access database and toolset to support access to compounds for virtual screening. The upgraded version ZINC^15^ makes it possible to carry out similarity searches and to explore the analogs of a given structure or part of a structure according to the input line employed [39]. The MTX structure in the SMILES (Simplified Molecular-Input Line-Entry System) format was used in ZINC^15^ to retrieve MTX-analog structures. The obtained analogs were filtered through ADMET and Lipinski’s Rule of five to categorize drug-like compounds. The validated pharmacophores WT-pharma and MT-pharma were then used as 3D-query to screen against drug-like MTX-analogs. The analogs mapped with WT- and MT-pharma were carried out for molecular docking where two compounds were found with a higher docking score than the reference (MTX). Further, molecular dynamics simulation confirmed one compound with a stronger affinity for WT and MT hDHFR yielding stable RMSD and strong molecular interactions with catalytic active site residues. Additionally, binding free energy calculation through MM/PBSA (Molecular Mechanics/Poisson-Boltzmann Surface Area) demonstrated robust binding affinity of Hit molecules with WT and MT hDHFR. Accordingly, in this study, we predicted an analog compound of MTX as a potential inhibitor for wild type and drug-resistant hDHFR for cancer therapeutics.

## 2. Results

### 2.1. Generation of Structures Based Pharmacophore Models

Crystal structures of wild type and F31R/Q35E variant of hDHFR in complex with methotrexate downloaded from the protein data bank were carried out for structure-based pharmacophore modeling. A total of 10 pharmacophore models for each structure, were generated while producing a ROC curve with each model. All the pharmacophores were attributed in terms of the total number of features, types of features, and selectivity score and ROC curve quality (Table 1). All ten models for wild type and for mutant structures yielded the same selectivity score with the difference in location of pharmacophoric features.

### 2.2. Pharmacophore Models Validation

Selected pharmacophore models termed as WT-pharma and MT-pharma were assessed for their sensitivity to retrieve the active compounds by receiver operating characteristics (ROC) curve. ROC curves were plotted with the generation of pharmacophore models by utilizing the option *Validation* that is available in the *receptor-ligand Phamacophore* module in DS for structure-based pharmacophore modeling. For this purpose, sets of 46 active and 24 inactive molecules were employed to testify model efficacy by creating the ROC curve. Higher the area under the ROC curve interpreted higher sensitivity of the model. For WT-pharma ROC displayed 0.989 and for MT-pharma 0.985 curve quality indicating 98.9% and 98.5% area under the curve illustrated as highly sensitive pharmacophore models to identify active molecules (Figure 2).

Additionally, Decoy set validation was implemented using a *Ligand Pharmacophore Mapping* module in DS. The accuracy of WT-pharma and Mt-pharma was evaluated by four factors i.e., false positive, false negative, enrichment factor (EF), and goodness of fit (GF). EF and GF were computed by applying the data of various parameters given in Table 2. Other properties of WT-pharma and MT-pharma including a percentage of the number of active yields (%Y), percent ratio of actives in the hit list (%A), false negatives, and false positives were also measured (Table 2).

### 2.3. Obtaining Methotrexate Analog Structures 

For the generation of structures analogous to methotrexate (MTX), we exploited MTX structure using SMILES format in ZINC^15^ for similarity search. Consequently, 32 compounds were retrieved (Appendix A. These compounds were downloaded in SDF format to visualize in DS and to carry out for further computations.

### 2.4. Drug-Likeness of MTX-analogs and Virtual Screening with Pharmacophore Models 

The compounds downloaded were subjected to ADMET and Lipinski’s Rule of five assessment tests to filter out drug-like MTX analogs. The ADMET assessment test gauged the pharmacokinetic features of the compounds obtained from ZINC^15^. In the ADMET assessment test, compounds were estimated for noninhibition to CYP2D6 and nonhepatotoxicity. The pharmacokinetic properties of blood brain barrier (BBB), optimal solubility, and good intestinal absorption were evaluated by setting their values to 3, 3, and 0, respectively. Lipinski’s rule of five assessment was carried out after ADMET evaluation. Through Lipinski’s rule of five filtration, compounds with AlogP value less than 5, number of HBD <5, number of HBA <10, molecular weight <500 Da, and fewer than 10 rotatable bonds were determined [40,41]. Accordingly, 8 compounds were found obeying drug-like properties. The drug-like MTX-analogs were carried out for virtual screening against WT-pharma and MT-pharma. All 8 compounds were aligned with WT-pharma but one compound was not in agreement with MT-pharma. Subsequently, 7 MTX-analogs were recognized as screening Hits for further computations. 

### 2.5. Molecular Docking of Screening Hits in Active Site of hDHFR 

To explore the binding mode of 7 drug-like compounds retrieved from virtual screening against WT-pharma and MT-pharma, molecular docking simulations were carried out using GOLD v 5.2.2. The 3D structure of wild type and F31R/Q35E variant of hDHFR in complex with inhibitor MTX were taken from protein data bank (PDB ID: 1U72 and 3EIG respectively). Both the structures have a high resolution of 1.9 Å for wild type and 1.7 Å for the F31R/Q35E variant. The co-crystal bound inhibitor (MTX) was docked in the active site of wild type hDHFR in order to optimize the docking protocol. The docked pose of MTX showed a low RMSD value of 0.58 Å with the crystallographic pose of MTX in the active site of the wild type hDHFR as shown in Appendix A. The WT- and MT-pharma retrieved drug-like (candidate) compounds were docked by implementing the same optimized protocol. Docking results showed that ChemPLP fitness scores and ASP fitness scores of MTX as a reference compound were 99.23 and 56.65 for wild type hDHFR while 88.98 and 49.84 for mutant hDHFR, respectively. These docking scores were considered as cut-off values for wild type and mutant hDHFR docking results analysis. The candidate compounds for wild type hDHFR were selected based on ChemPLP and ASP fitness scores greater than 99.23 and 56.65 respectively. For mutant hDHFR, compounds yielding ChemPLP and ASP fitness scores higher than 88.98 and 49.84 were selected (Table 3). 

Additionally, the compounds were investigated about ligand conformations effectively showing essential interactions in the active site of hDHFR. Finally, one compound which contained pharmacophoric features of wild type and mutant hDHFR structures and fulfilled the above-mentioned criteria of docking scores was selected as docking Hit. The pharmacophore mapping of Hit compound (ID: ZINC000013508844) with WT-pharma and MT-pharma models are shown (Figure 3).

### 2.6. Molecular Dynamic Simulations for Structures Stability Evaluation

MD simulations were executed to estimate the binding stability of Hit compound after docking in the active site of wild type and mutant hDHFR. Four MD simulation systems were employed as one for each complex i.e., for hit compound and reference compound (MTX) in complex with wild type and mutant hDHFR structures, respectively. The initial details of each system subjected to simulation are given in Table 4. 

Root mean square deviation (RMSD) was measured of the protein-ligand complex for each simulation system to assess ligand binding with hDHFR. In the results of 50 ns simulation, protein-ligand RMSD of reference compound (MTX) in complex with wild type hDHFR was recorded at an average of 0.21 nm throughout the simulation (Figure 4A). The average RMSD of MTX with mutant hDHFR was observed 0.21 nm up to 38.9 ns but afterward it significantly increased to an average of 0.62 nm indicating loss of MTX binding with MT-hDHFR. Accordingly, the representative structure of each system was taken from the last 8 ns (30–38 ns) before the loss of MTX binding with MT-hDHFR. The Hit compound obtained from docking results showed stable RMSD in complex with WT- and MT- hDHFR. The average root means square deviation values of Hit compound in complex with WT-hDHFR and MT-hDHFR were observed at an average of 0.21 nm and 0.22 nm respectively, throughout the simulation depicting that both the systems were well converged (Figure 4A). Additionally, per residue RMSF (root mean square fluctuation) calculated for each complex which was noted about 0.3 nm for all residues except for the MTX which showed RMSF about 2.3 nm in complex with MT-hDHFR (Figure 4B).

The superimposition of representative structures demonstrated that the binding pattern and conformational alignment of Hit in the active site of hDHFR was similar to that of MTX (Figure 5).

The substrate-binding site of hDHFR is mainly comprised of Ile7, Glu30, Phe31, Gln35, Ser59, Pro61, Asn64, Arg70 and Val115 [42]. Our results suggested that the reference compound (MTX) could bind with substrate binding residues of WT-hDHFR but lost its binding with MT-hDHFR, in agreement with Volpato et al. [32]. In contrast with MTX, the Hit compound exhibited strong binding with the active site of both WT- and MT-hDHFR. The MTX formed H-bonds with Ile7, Glu30, and Val115, Phe31, Asn64 and Arg70, as well as one carbon–hydrogen with Pro61 of WT-hDHFR (Figure 6A, Table 5). Furthermore, MTX established π-interactions with Ile7, Ala9, Leu22, Phe34 and Ile60 and showed van der Waals contacts with Val8, Asp21, Phe31, Arg32, Tyr33, Gln35, Thr56, Ser59, Leu67, Lys68, Tyr121 and Thr136 (Table 5).

In the representative structure of MT-hDHFR which was obtained before the disruption of MTX binding, molecular interactions were observed to analyze the difference of MTX binding with wild type hDHFR, leading us to speculate about the segment of resistance to MTX in mutant hDHFR. Accordingly, MTX was shown to form H-bond interactions with Ile7, Glu30, Arg31, Asn64, Lys68, Val115, Tyr121 and van der Waals interaction with Asp21, Phe34, Tyr33, Glu35, Thr56, Pro61, Arg70, Phe134 and Thr136 (Figure 6C, Table 5). Our findings also indicate that MTX formed carbon–hydrogen bonds with Val8, Leu67, Ser59, Lys68 and π-interactions with Asp21, Phe34, Tyr33, Glu35, Thr56, Pro61, Arg70, Phe134 and Thr136. The Hit compound in complex with WT-hDHFR formed H-bonds with Ile7, Glu30, Gln35, Asn64 (2), Arg70 and Val115, as well as carbon–hydrogen bonds with Pro61 and Lys68 (Figure 6B, Table 5). Additionally, Hit showed van der Waals interactions with the hydrophobic pocket residues of WT-hDHFR such as Val8, Asp21, Phe31, Tyr33, Phe34, Thr56, Ser59, Leu67 and Thr136, as well as π-interactions with Ile7, Ala9, Leu22 and Ile60 (Table 5). In the case of MT-hDHFR, the Hit compound established H-bonds with Ile7, Glu30, Arg31, Ser59, Asn64, Arg70, Val115 and Tyr121 (Figure 6D, Table 5). The Hit compound showed hydrophobic van der Waals interactions with Val8, Asp21, Arg28, Arg32, Phe34, Glu35, Thr56, Pro61, Leu67 and Thr136 residues of the WT-hDHFR while π-interactions were formed with Ile7, Ala9, Leu22, Arg31 and Ile60. The residue Ser59 also exhibited carbon–hydrogen bonds with C11 atoms in addition to conventional H-bonds with O13 atoms in Hit molecules. The conventional H-bond was formed only by Hit in the MT-hDHFR binding site. Throughout the simulation period, the total number of intermolecular H-bonds of WT- and MT- hDHFR in complex with MTX and Hit were calculated. Our results showed that the Hit compound formed an average number of H-bonds with WT- and MT-hDHFR comparable to that of MTX (reference) in WT-hDHFR. Since MTX has a very weak binding with MT-hDHFR, it could not maintain average number of H-bonds after 38.9 ns (Figure 4C), which enhanced the results obtained from the RMSD plots. Our results suggest that Hit (MTX analog) is capable of binding tightly with wild type, as well as MTX-resistant, F31R/Q35E hDHFR variants. 

### 2.7. Binding Free Energy Calculations for MTX and Hit Compound 

MM/PBSA binding free energies were calculated for MTX and Hit in complex with WT- and MT-hDHFR. The free energies of MTX and Hit in complex with WT-hDHFR were observed as −646.76 kJ/mol and −642.12 kJ/mol, whereas MT-hDHFR MTX could yield only −49 kJ/mol, while the Hit compound attained −571.38 kJ/mol. The binding free energy evaluations underscore our findings that the Hit molecule is tightly bound with WT- and MT-hDHFR, displaying comparable free energy of MTX in complex with WT-hDHFR. The decomposition analysis of the binding free energy indicated that electrostatic and van der Waals forces are significant characteristics in hDHFR inhibition (Figure 7, Table 6).

Altogether, our results verified that the newly identified MTX-analog favorably adapted the active site of wild type and double mutant hDHFR and acquired polar and nonpolar interactions with the catalytic active residues.

The structure of the Hit compound, which was modified by adding a carbon-oxygen group (C11-C12-O13) with a *p*-ABA moiety, is illustrated in its 2D structure in Figure 8. 

## 3. Discussion

Chemotherapeutics are very effective in the treatment of cancers, but drug resistance is often a limiting factor. Acquired resistance is the type of drug resistance that can develop through various adaptive responses such as mutations, increased expression of the therapeutic target and activation of alternative compensatory signaling pathways arising over the course of the treatment of tumors [43] Human dihydrofolate reductase (hDHFR) is an enzyme that is responsible for the catalysis of the reduction of 7,8-dihydrofolate (DHF) to 5,6,7,8-tetrahydrofolate, which is crucial for DNA synthesis and cell proliferation [44]. Therefore, hDHFR has been widely used as a target for cancer therapeutics for several decades [45]. Methotrexate is a well-known inhibitor that displays a high affinity with hDHFR, but mutation in the active site of hDHFR results in the loss of MTX binding [31].

The present study aimed to identify an analog of methotrexate that was capable of binding tightly, and hence inhibiting, wild type and doubly mutant hDHFR (F31R/Q/35E) by employing several computational methods including structure-based pharmacophore modeling, virtual screening, molecular docking and molecular dynamics simulations studies [46]. Structure-based pharmacophore models of crystal structures of wild type (PDB ID: 1U72) and variant (PDB ID: 3EIG) hDHFR in complex with methotrexate were obtained, with four features in each model. The best pharmacophore models of each the structure were selected by analyzing the inclusion of key residues in pharmacophoric features and sensitivity of the models to retrieve true positive compounds depicted by the highest area under the ROC curve. The selected pharmacophore models, termed as WT-pharma and MT-pharma for wild type and mutant hDHFR structures, respectively, were rationally assessed for the inclusion of conserved hydrogen bond residue Glu30 and other key residues, such as Asn64, Arg70 and Val115 [44,47]. Further, with each pharmacophore model, the ROC curve was formed between the number of false positive (FP) and true positive (TP) compounds retrieved by that model from the datasets of 46 active and 24 inactive compounds. Higher AUC values in the ROC curves infer greater sensitivity of WT-pharma and MT-Pharma in retrieving actives, and specificity for ignoring inactives [48,49]. Using ZINC^15^, Mayorga et al. found a high number of compounds when they utilized a small fragment of the original structure [50]. In our study, to explore analogs of MTX, we used the full structure of MTX and selected ‘in vitro’, ‘in vivo ‘and ‘clean’ options in the section of ‘subsets’, which resulted in the generation of only 32 analogous compounds (Appendix A. An ADMET assessment test and Lipinski’s Rule of five scrutinized the downloaded compounds from ZINC^15^ for their drug-like properties, and found eight compounds satisfying the required criteria to qualify as lead compounds [51,52]. The validated pharmacophore models of wild type and mutant structures of hDHFR were applied as 3D query for virtual screening with the drug like compounds capable of binding with wild type and mutant hDHFR as well [53]. A molecular docking study was employed to inspect the most suitable and consistent binding mode of the molecules in the binding sites of receptor proteins. Consequently, the best binding modes obtained from docking based on scoring functions and key interactions with the active site residues of wild type and mutant hDHFR were used in MD simulations to assess their stability [54]. The RMSD plots inferred that the Hit compound showed similar modes of interaction in wild type and mutant hDHFR active sites as MTX in the active site of wild type hDHFR. Specifically, the average RMSD profiles (<0.25 nm) obtained for protein-ligand complexes of Hit with wild type and mutant hDHFR exhibited that the systems were uniform and compact, as the stability of the system can be inferred by an RMSD value of less than 0.3 nm [40,41]. The RMSD plot for MTX in complex with mutant hDHFR showed abrupt fluctuation after 38.9 ns, which indicated the loss of MTX binding with the active site of hDHFR. Furthermore, a high RMSF value (2.34 nm) of MTX (residue187) indicated a loss of ligand binding with only mutant hDHFR protein. Our results showed that Hit compound established stable H-bonds with the active site residues of wild type and mutant hDHFR. Similar to the molecular interactions of MTX (reference compound), most H-bonds were formed by pterin moiety and α-glutamate moiety with hDHFR active site residues, while *p*-aminobenzoic acid (*p*-ABA) moiety formed mainly hydrophobic interactions [31]. The conserved hydrogen bond with OE1 atom of catalytic residue Glu30 was formed with the pterin moiety of Hit molecule [42,55]. The additional oxygen atom in the structure of Hit compound formed a hydrogen bond with Ser59 in both wild type and mutant hDHFR, while Ser59 belonged to the coenzyme NADPH binding site [56]. It was speculated that the hydrogen bond between Ser59 and the modified *p*-ABA moiety of Hit compound contributed to the strong binding of the Hit compound with the mutant structure of hDHFR. Furthermore, the conserved hydrogen bonds formed by an α-carboxylate group of MTX with the side chains of Arg70 and Gln35 while *p*-aminobenzoyl keto group with Asn64 were also observed in Hit compound’s interactions with wild type hDHFR [26]. 

In the mutant structure, due to the substitution of Glu35, a hydrogen bond was not formed because of unfavorable close electrostatic contact of two negative charges between Glu35 side chain and glutamate moiety of MTX and Hit compound. In contrast, Arg31, which was substituted at the position of Phe31, was observed to form hydrogen bonds through the guanidinium group with an α-glutamate moiety of MTX. Hit compound displayed a double hydrogen bond with Arg31; double hydrogen bonds are considered to be crucial for strong binding in protein-ligand interactions [57]. Furthermore, the hydrogen bond of the Arg70 side-chain with an α-carboxylate group of MTX was lost in Mt hDHFR, while Lys68 was formed a hydrogen bond with the α-carboxylate group. On the other hand, the Hit compound retained the H-bond of the Arg70 side chain with the α-carboxylate group, as it did in the wild type. Moreover, the only hydrogen bond formed by a *p*-ABA group of MTX with Asn64 in wild type hDHFR was also shifted to a H-bond with an α-carboxylate group of MTX in mutant hDHFR. While the *p*-ABA group of Hit compound formed a hydrogen bond with Asn64, as in the wild type, an additional H-bond was formed between Ser59 and modified oxygen atom added to *p*-ABA group. Accordingly, a comparative analysis of protein-ligand interactions of MTX and Hit compound suggested that Hit (MTX-analog) may be capable of retaining its strong binding with WT and MT hDHFR. Additionally, the binding free energy evaluations performed by the MM/PBSA method also inferred that the complexes of WT and MT hDHFR with Hit compound were comparably stable, like MTX in WT hDHFR; meanwhile, the binding free energy profile noticeably depicts the loss of MTX binding in MT hDHFR.

## 4. Materials and Methods

### 4.1. Structure Based Pharmacophore Modeling

Ligand binding features were assessed by the structures of wild type (PDB ID: 1U72) and drug-resistant (PDB ID: 3EIG) human DHFR in complex with methotrexate taken from protein data bank [26,31]. Pharmacophore models were generated using the *Receptor-ligand Pharmacophore Generation* module in Discovery Studio (DS) v.4.5 (Dassault System, BIOVIA Corp, San Diego, CA, USA). FAST (Features from Accelerated Segment Test) algorithm was applied for Conformation Generation, while the Fitting Method was set to Flexible. The Validation option was set to *True*, for which a set of 46 active and 24 inactive compounds, downloaded from BindingDB (https://www.bindingdb.org/bind/index.jsp) were exploited to generate a ROC curve for each pharmacophore model.

### 4.2. Decoy Test Validation

The ability of pharmacophore to identify hDHFR inhibitors was assessed by the Guner−Henry method (Decoy test method) [44]. A test set was prepared by collecting hDHFR inhibitors whose experimental activities (IC_50_ values) were measured by the same biological assays. The test set was composed of active and inactive molecules of hDHFR. The selected pharmacophore models of wild type and mutant structures were employed as a 3D query to obtain the best-fitted molecules from the test set. Screening of the test set was executed by the Ligand Pharmacophore Mapping protocol embedded in DS. Accordingly, several parameters, like Guner−Henry (GH) score, enrichment factor (EF), the percentage ratio of actives (%A), percentage yield of actives (%Y), false negative and false-positive, were calculated, which determined the efficacy of WT-pharma and MT-pharma
EF = (H_a_/H_t_)**/**(A**/**D)GF = (H_a_/4H_t_A) (3A + H_t_) × [{1 − (H_t_ − H_a_)**/**(D − A)}](1)
where D is the total molecules in the data set, A specifies the total number of active compounds in the data set, H_t_ indicates the total number of Hits retrieved and H_a_ refers to the number of actives present in the retrieved Hits.

### 4.3. Methotrexate Analogs Generation

Methotrexate structure was subjected to a similarity search in ZINC^15^ using SMILES string of MTX. For the generation of clean analogs*,* in vivo and in vitro options were selected in the available range of Subsets to Check. Subsequently, the structures were downloaded in the SDF (Spatial Data File) format, generated by the webserver, to carry out for further computations in DS.

### 4.4. Drug-Likeness Prediction and Virtual Screening

The molecules retrieved from ZINC^15^ were tested through ADMET and Lipinski’s Rule of five embedded assessment techniques in DS to identify drug-like compounds. Subsequently, the compounds exhibiting such properties were carried out for virtual screening with WT-pharma and MT-pharma. The compounds which fitted with both pharmacophores were considered as screening compounds in our molecular docking study.

### 4.5. Molecular Docking Simulation

A docking study was employed through the Genetic Optimization of Ligand Docking (GOLD) package v5.2.2 (The Cambridge Crystallographic Data Centre, Cambridge, United Kingdom). GOLD software provides full flexibility of ligands and limited flexibility of protein; hence, it delivers more reliable results in computational biology the crystal structures of wild type (PDB ID: 1U72) and variant (PDB ID: 3EIG) hDHFR in complex with Methotrexate were taken from protein data bank. The wild type and variant structures of hDHFR were prepared for docking by eliminating water molecules in DS. Chemistry at Harvard macromolecular mechanisms (CHARMm) force field was applied to add hydrogen atoms to the structures of hDHFR. The binding sites of wild type and mutant hDHFR were identified within the radius of 9Å of bound inhibitor (MTX) using the *Define and Edit Binding Site* module, planted in DS. During docking, MTX-analogs retrieved from virtual screening along with methotrexate as reference were treated as ligand molecules. The ChemPLP (Piecewise Linear Potential) score and ASP (Astex Statistical Potential) score were used as the default scoring and rescoring functions, respectively. The ChemPLP is the default scoring function in GOLD software which is empirically optimized for pose prediction. It is implemented to establish the steric complementarity between protein and ligand, distance- and angle-dependent hydrogen and metal bonding terms as well as the heavy atoms clash- and torsional potential. The ASP scoring function measures the atom−atom potential and has similar precision to Chemscore and Goldscore fitness functions [58,59]. During docking, GA (genetic algorithm) was run to produce 100 poses for each drug-like molecule. The bound ligand (MTX) was employed as a reference compound throughout the analyses. Cluster analysis was performed to scrutinize hit compounds exhibiting a higher dock score than cut off (dock score of reference molecule).

### 4.6. Molecular Dynamics (MD) Simulation

Molecular dynamics simulations were performed using CHARMm36 all-atom force field [60] in Groningen Machine for Chemical Simulation (GROMACS) v5.0.6 package [61]. For every protein-ligand complex, an independent simulation system was generated. The topology and coordinates files for MTX and docking hits were generated using SwissParam [62]. Transferable intermolecular potential with three points (TIP3P) water model in a cubic box was used for solvation of each system. Solvent molecules were substituted with sodium ions (Na^+^) to nullify the total charge of simulation systems. The energies of the systems were minimized by applying steepest decent algorithwhere the maximum force was kept less than 10 kJ/mol I order to avoid any bad contacts likely to be occurred in the production run. Initially, the systems were equilibrated in two steps. First, the number of particles at constant volume and temperature (NVT) equilibration was carried out for 100 ps at 300 K. The temperature of the system was kept constant using V-rescale thermostat [63]. In second phase, 100 ps equilibration was performed with number of particles at constant pressure of 1 bar (NPT) and temperature 300 K [64]. Accordingly following the protocol mentioned earlier, all the systems were carried out for production run. In short, bond lengths of heavy atoms were sustained using Linear Constraint Solver (LINCS) algorithm [65]. Particle Mesh Ewald (PME) method was employed to calculate the long-range electrostatic interactions [66]. Short-range interactions length was kept to 12 Å. All simulations were performed with the periodic boundary conditions to make infinite systems. Time interval was kept of 10 ps to save coordinates data. Finally, result’s visualization and analysis were performed using GROMACS and DS.

### 4.7. Binding Free Energy Calculations

The binding free energy was calculated by employing the Molecular Mechanics/Poisson-Boltzmann Surface Area (MM/PBSA) method [67]. Following the MM/PBSA protocol to compute free energies of the protein-ligand complex, equidistant snapshots of the hDHFR-ligand complex were extracted. The binding free energy of the protein-ligand complex is stated as:Δ𝐺_binding_ = 𝐺_complex_ − (𝐺_protein_ + 𝐺_ligand_)(2)
where 𝐺_complex_ denotes the sum of the free energy of the complex, and 𝐺_protein_ and 𝐺_ligand_ specify the free energies of portion and ligand in their unbound states.

Free energy can be defined as:𝐺_𝑋_ = 𝐸_MM_ + 𝐺_solvation_(3)
where 𝑋 can be a protein, a ligand, or their complex. 𝐸_MM_ signifies the average molecular mechanics potential energy in vacuum, while 𝐺_solvation_ indicates the free energy of solvation.

Accordingly, molecular mechanics potential energy in vacuum can be calculated by implementing the equation:𝐸_MM_ = 𝐸_bonded_ + 𝐸_nonbonded_ = 𝐸_bonded_ + (𝐸_vdw_ + 𝐸_elec_)(4)

𝐸_bonded_ denotes the bonded interactions, while 𝐸_nonbonded_ terms the nonbonded interactions. The value of Δ𝐸_bonded_ is generally treated as zero.

The combined energetic terms of electrostatic (𝐺_polar_) and apolar (𝐺_nonpolar_) give the solvation free energy which is measured as: 𝐺_solvation_ = 𝐺_polar_ + 𝐺_nonpolar_(5)

Here Poisson-Boltzmann (PB) equation is implemented to compute G_polar_, while the 𝐺_nonpolar_ is calculated from the solvent-accessible surface area (SASA) as:𝐺_nonpolar_ = γ^SASA^ + 𝑏(6)
where, γ represents the coefficient of solvent surface tension, while 𝑏 is its fitting parameter, whose values are 0.02267 kJ/mol/Å^2^ and 3.849 kJ/mol, respectively.

## 5. Conclusions

In the current study, structure-based pharmacophore modeling, virtual screening, molecular docking and molecular dynamics simulation methods were utilized to identify a potential inhibitor that was capable of strong binding with wild type as well as drug-resistant (mutant) hDHFR. Structure-based pharmacophore models for WT and MT hDHFR in complex with MTX were generated and validated by the decoy test and ROC curve. Methotrexate analogs were generated by exploiting the MTX structure in ZINC^15^, and carried out for ADMET and Lipinski’s Rule of five assessment tests to evaluate drug-likeness of compounds obtained from ZINC. The drug-like compounds were used in virtual screening with validated WT and MT pharmacophore models as a 3D query to identify potential hits of wild type and mutant hDHFR. The compounds obtained from virtual screening were docked in the active site sites of WT and MT hDHFR. Subsequently, through docking results analysis, one compound was found to have a higher dock score than the reference compound (MTX), displaying essential molecular interactions with key residues of the hDHFR active site. Furthermore, MD simulation and binding free energy calculations for the Hit compound and MTX in complex with WT and MT hDHFR were also used to evaluate the stability of the Hit compound with WT and MT hDHFR. Taken together, our findings indicate MTX analog (ZINC000013508844) to be a potential inhibitor of wild type hDHFR and drug-resistant F31R/Q35E variant of hDHFR. In future work, we will try to synthesize the Hit compound to verify our findings through bioassay by collaborating with an experimental lab. These findings can also be extended to assess other drug resistant hDHFR variants for cancer therapeutics.

## Figures and Tables

**Figure 1 molecules-25-03510-f001:**
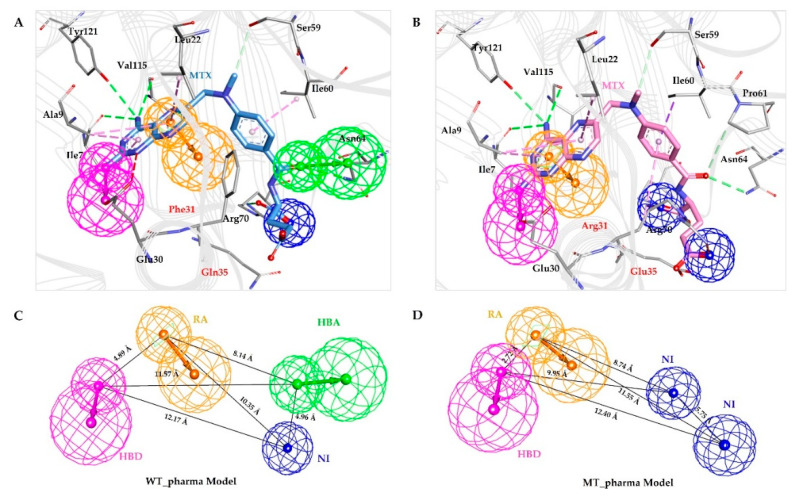
Structure-based pharmacophore generation. (**A**) Residues of wild type hDHFR active site complementing pharmacophoric features are shown as a thin stick. Bound inhibitor (MTX) is shown as a light blue colored thick stick model. HBA, HBD, RA, and NI are colored as green, magenta, orange and blue respectively. (**B**) Residues of mutant hDHFR active site complementing pharmacophoric features are shown as a thin stick. Bound inhibitor (MTX) is shown as a pink-colored thick stick model. HBA, HBD, RA, and NI are colored as green, magenta, orange and blue respectively. (**C**) Interfeature distance illustration of WT-pharma (**D**) Interfeature distance illustration of MT-pharma.

**Figure 2 molecules-25-03510-f002:**
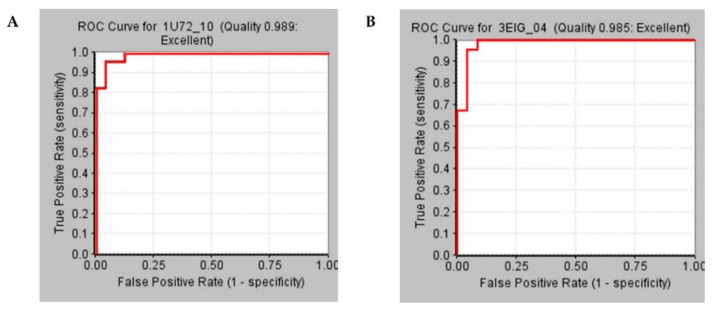
Receiver Operating Characteristics curves for validation of selected pharmacophore models between true positive and false-positive rates. (**A**) ROC curve shown in the red line for the WT-pharma model with 0.989 curve quality depicts 98.9% area under the curve. (**B**) ROC curve shown in the red line for the MT-pharma model with 0.985 curve quality depicts 98.5% area under the curve.

**Figure 3 molecules-25-03510-f003:**
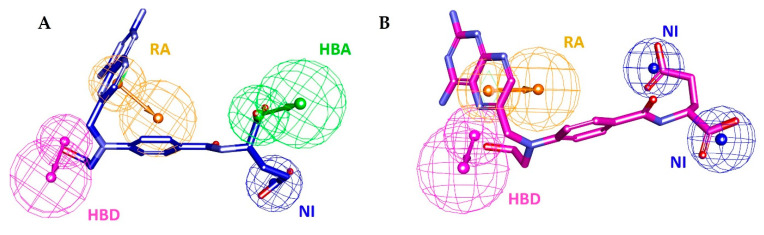
Hit compound (MTX-analog) mapping with pharmacophore models. (**A**) Hit compound represented in dark blue colored thick stick model mapping with WT-pharma. (**B**) Hit compound represented in magenta-colored thick stick model mapping with MT-pharma.

**Figure 4 molecules-25-03510-f004:**
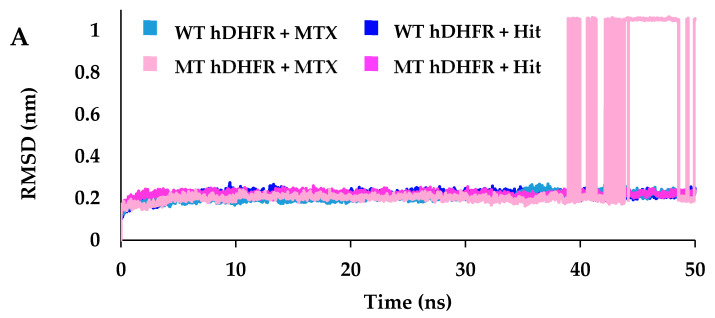
RMSD analysis of the reference (MTX) and hit compound (MTX-analog). (**A**) RMSD of the protein-ligand complex of wild type and mutant hDHFR revealed their stability throughout the simulation, with no abnormal behavior in all systems except for MTX in complex with MT hDHFR. (**B**) RMSF per residue plot for all the systems portrayed their residues RMSD is stable except for MT hDHFR ligand (MTX) which showed a high fluctuation level. (**C**) The number of intermolecular hydrogen bonds between protein and ligand during 50 ns MD simulations. Light blue and pink colors represent MTX in wild type and mutant hDHFR, respectively, while dark blue and magenta represent the Hit compound in wild type and mutant hDHFR, respectively.

**Figure 5 molecules-25-03510-f005:**
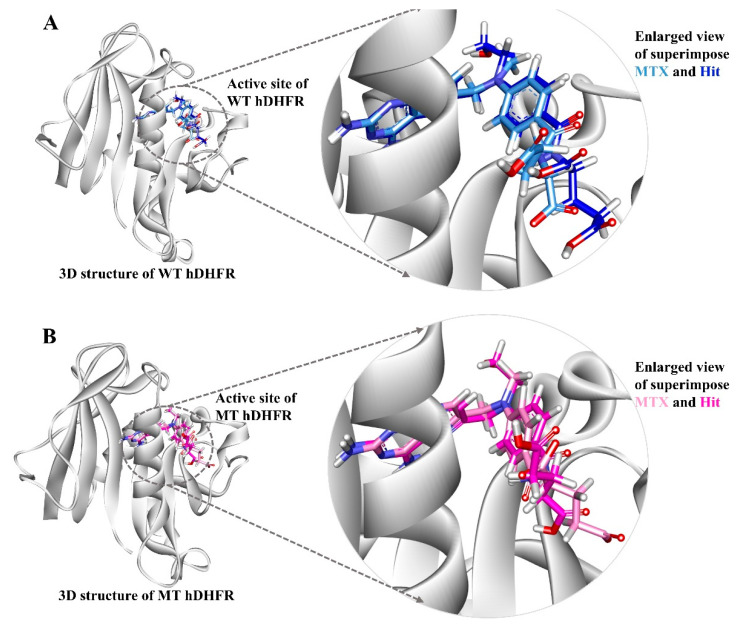
The binding patterns of the reference inhibitor (MTX) and hit compound in the active site of wild type and mutant hDHFR. Compounds are displayed by their representative structures superimposed (left) and enlarged (right). The protein is shown in white color. (**A**) Light blue and dark blue colors represent MTX and Hit compound in wild type hDHFR. (**B**) Pink and magenta colors represent MTX and Hit compound respectively in mutant hDHFR.

**Figure 6 molecules-25-03510-f006:**
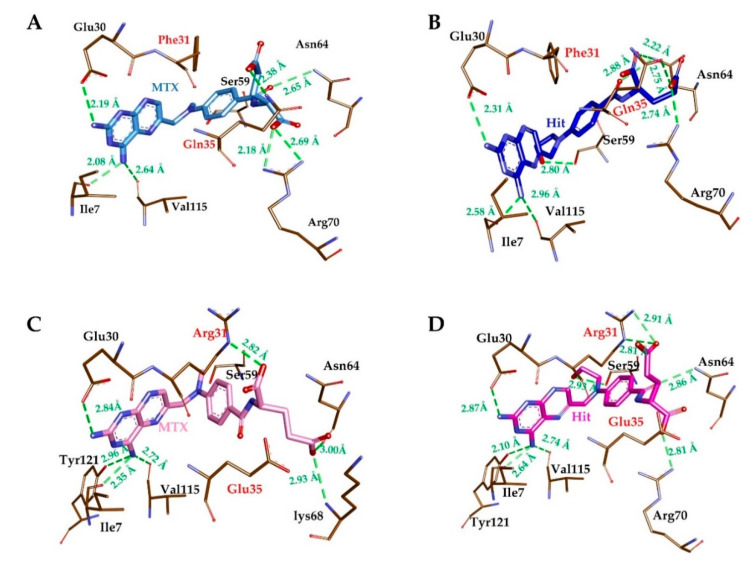
Molecular interactions analyses. The reference inhibitor MTX and Hit compound interacted with essential residues in the active site of hDHFR. MTX in WT hDHFR (**A**), Hit in WT hDHFR (**B**), MTX in MT hDHFR (**C**) and Hit in MT hDHFR (**D**) are depicted as light blue, dark blue, pink, and magenta-colored stick representation. The H-bond forming residues of hDHFR are displayed as a brown stick model. H-bonding and bond distances are represented as green dashed lines and measured in angstrom (Å), respectively.

**Figure 7 molecules-25-03510-f007:**
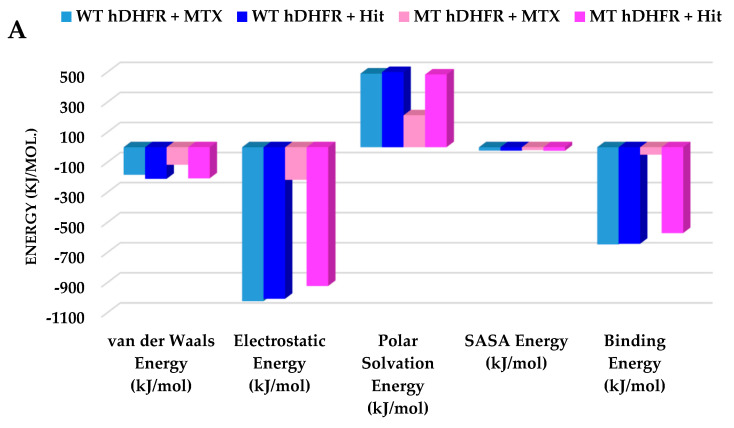
Binding free energy analyses. (**A**) Graphical representation of MM/PBSA estimated binding free energy of wild type and mutant hDHFR in complex with MTX (reference) and Hit compound throughout the simulation time. The reference compound is depicted as light blue and dark blue for wild type and mutant hDHFR, respectively. The Hit compound is shown in pink and magenta colors for wild type and mutant hDHFR, respectively. (**B**) The binding free energy decomposition analysis of the final hits in the active site of hDHFR infers that the Hit compound was comparably strongly bound with WT and MT hDHFR, while MTX lost its binding with the mutant structure.

**Figure 8 molecules-25-03510-f008:**
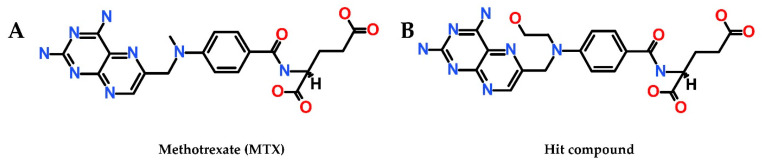
(**A**) 2D structure of MTX (**B**) 2D structure of Hit compound (MTX analog, ZINC ID: ZINC000013508844).

**Table 1 molecules-25-03510-t001:** Receptor-ligand based pharmacophores characteristics details.

Sr. No.	Number of Features	WT hDHFR Phrmacophore Details			MT hDHFR Phrmacophore Details		
		Features Set	Selectivity Score	ROC Curve Quality	Features Set	Selectivity Score	ROC Curve Quality
Pharmacophore_1	4	HBD, HBD, HYP, NI	11.090	0.832	HBD, HBD, NI, NI	12.455	0.944
Pharmacophore_2	4	HBD, HBD, NI, RA	11.090	0.924	HBA, HBD, NI, NI	12.455	0.929
Pharmacophore_3	4	HBD, HBD, HYP, NI	11.090	0.886	HBA, HBD, NI, NI	12.455	0.913
Pharmacophore_4	4	HBD, HBD, NI, RA	11.090	0.951	HBD, NI, NI, RA	12.455	0.985
Pharmacophore_5	4	HBD, HBD, NI, RA	11.090	0.903	HBD, NI, NI, RA	12.455	0.968
Pharmacophore_6	4	HBA, HBD, HBD, NI	11.090	0.941	HBD, HYP, NI, NI	12.455	0.946
Pharmacophore_7	4	HBA, HBD, NI, RA	11.090	0.937	HBD, HYP, NI, NI	12.455	0.955
Pharmacophore_8	4	HBD, HYP, NI, RA	11.090	0.822	HBD, NI, NI, RA	12.455	0.958
Pharmacophore_9	4	HBD, HYP, NI, RA	11.090	0.907	HBD, NI, NI, RA	12.455	0.962
Pharmacophore_10	4	HBA, HBD, NI, RA	11.090	0.989	HBD, HYP, NI, NI	12.455	0.933

The pharmacophore models comprising of features including key residues Glu30, Asn64, Arg70, and Val115 were selected so-called WT-pharma and Mt-pharma for wild type and mutant structures respectively. WT-pharma consisting of four pharmacophoric features included one hydrogen bond acceptor (HBA), one hydrogen bond donor (HDB), one negative ionizable (NI) and one ring aromatic (RA). MT-pharma comprised of features one hydrogen bond donor (HDB), two negative ionizable (NI) and one ring aromatic (RA) (Figure 1).

**Table 2 molecules-25-03510-t002:** Decoy set validation for WT & MT hDHFR structure-based pharmacophore models. WT-pharma and MT-pharma obtained the highest goodness of fit score suggesting the suitability of the models for virtual screening.

Parameters	Values (WT hDHFR)	Values (MT hDHFR)
Total no. of molecules in the database (D)	90	90
Total no. of actives in the database (A)	20	20
Total no. of hit molecules from the database (H_t_)	25	17
Total no. of active molecules in hit list (H_a_)	19	17
Percentage Yield of actives [(H_a_/H_t_) × 100]	76%	100%
Percentage Ratio of actives [(H_a_/A) × 100]	95%	85%
Enrichment Factor [EF = (H_a_/H_t_)/(A/D)]	3.4	4.5
False negatives (A − H_a_)	1	13
False positive (H_t_ − H_a_)	6	0
Goodness of fit score [GF = (H_a_/4H_t_A)(3A + H_t_) × [{1 − (H_t_ − H_a_)/(D − A)}]]	0.93	0.96

**Table 3 molecules-25-03510-t003:** Comparison of ChemPLP and ASP dock scores of MTX (reference inhibitor) and Hit compound in the active sites of WT and MT hDHFR.

System	ChemPLP Score	ASP Score
WT hDHFR + MTX	99.23	56.65
WT hDHFR + Hit	103.74	57.70
MT hDHFR + MTX	88.98	49.84
MT hDHFR + Hit	91.07	47.59

**Table 4 molecules-25-03510-t004:** The specifications of four systems used for molecular dynamics simulations.

System	No. of TIP3P Water Molecules	No. of Na^+^ Ions	System Size (nm)
WT hDHFR + MTX ^a^	7726	1	7.11 × 7.11 × 5.03
WT hDHFR + Hit	7646	1	7.11 × 7.11 × 5.03
MT hDHFR + MTX	8258	2	7.11 × 7.11 × 5.03
MT hDHFR + Hit	8181	1	7.11 × 7.11 × 5.03

MTX ^a^: the reference inhibitor.

**Table 5 molecules-25-03510-t005:** Molecular interactions between the ligands (MTX and hit compound) and the active site residues of WT and MT hDHFR.

Compound	Hydrogen Bond Residues (<3Å)	van der Waals Residues	Carbon Hydrogen Bond Residues	π-Interaction Residues
MTX (with WT hDHFR)	Ile7, Glu30, Asn64, Arg70(2), Gln35, Val115	Val8, Asp21, Phe31, Arg32, Tyr33, Thr56, Ser59, Leu67, Lys68, Tyr121, Thr136	Pro61	Ile7, Ala9, Leu22, Phe34, Ile60
Hit (with WT hDHFR)	Ile7, Glu30, Gln35, Ser59, Asn64(2), Arg70, Val115	Val8, Asp21, Phe31, Tyr33, Phe34, Thr56, Leu67, Thr136	Pro61, Lys68	Ile7, Ala9, Leu22, Ile60
MTX (with MT hDHFR)	Ile7, Glu30, Arg31, Asn64, Lys68, Val115, Tyr121	Asp21, Phe34, Tyr33, Glu35, Thr56, Pro61, Arg70, Phe134, Thr136	Val8, Leu67, Ser59, Lys68	Ile7, Ala9, Leu22, Arg31, Ile60
Hit (with MT hDHFR)	Ile7, Glu30, Arg31 (2), Ser59, Asn64, Arg70, Val115, Tyr121	Val8, Asp21, Arg28, Arg32, Phe34, Glu35, Thr56, Pro61,Leu67, Thr136	Ser59	Ile7, Ala9, Leu22, Arg31, Ile60

**Table 6 molecules-25-03510-t006:** Decomposition of binding free energy.

Complex	Van der Waals Energy (kJ/mol)	Electrostatic Energy (kJ/mol)	Polar Solvation Energy (kJ/mol)	SASA ^b^ Energy (kJ/mol)	Binding Energy (kJ/mol)
WT hDHFR + ^a^MTX	−184.057	−1023.945	489.982	−22.594	−646.767
WT hDHFR + Hit	−210.358	−1007.98	499.622	−22.622	−642.123
MT hDHFR + MTX	−116.884	−217.191	212.294	−18.862	−49.299
MT hDHFR + Hit	−207.152	−923.188	483.648	−23.977	−571.381

^a^ MTX: methotrexate as reference inhibitor. SASA ^b^: Solvent accessible surface area.

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
