# Peer review of "In Silico Study Identified Methotrexate Analog as Potential Inhibitor of Drug Resistant Human Dihydrofolate Reductase for Cancer Therapeutics"

_molecules, 2020, doi:10.3390/molecules25153510_

Round 1

Reviewer 1 Report

The manuscript entitled "In Silico Study Identified Methotrexate Analog as Potential Inhibitor of Drug-Resistant Human Dihydrofolate Reductase for Cancer Therapeutics" Rana and Colleagues were trying to identify analog of methotrexate capable of binding and inhibiting wild type and double mutant hDHFR (F31R/Q/35E) by employing several computational methods. The manuscript is easy to read and understand the experiments that were done.

Some typing errors are present i.e.

-In the abstract, row 34, "Undergoing through clustering analysis and protein-ligand interactions scrutiny, one compounds was found with.."

-In conclusion, row 524 "In the current study.."

Reviewer 2 Report

The authors presented an extensive in silico study on methotrexate and its derivatives as a potential compounds to overcome drug resistance in cancer treatment. Although, many aspects have been touched in the study, and the manuscript is very comprehrensive in terms of the computational methods used, there is one problem, which according to my opinion, makes the study too premature for publication - there is no experimental validation of the in silico results, and all conclusions drawn from the study are based on speculations. Without any experimental validation of the results obtained in the computational part of the study, I cannot accept the manuscript for publication.

Reviewer 3 Report

The authors present a complete study, focussing on many relevant aspects and using appropriate tools. The ROC curves reveal a high performance of the suggested model.

 I feel that the article can be published in its present form. The only aspects to be formally corrected are:

-In Figure 4C some lines overlap with other and hide the information. Please, clean it if possible, or present four subgraphs.

-There is no Figure 5.

-In Figure 8: structures A and B are almost the same (as said in text). Try to draw one as a pure copy of the other and adding the additional residue. In other words: draw exactly in the same manner (i.e. pure copy) the common skeleton (showing the same molecular conformation).

Round 2

Reviewer 2 Report

The authors do not plan extend the study by any experimental tests at this moment. Therefore, I stand by my decision and do not recommend publication at this moment.

Author Response

We appreciate reviewer's high concerns. Since our lab is not equipped for wet lab experiments so we are trying to collaborate with some other lab to perform biological assays. That's why we are sorry that cannot provide any experimental tests at this stage. Also our plan to extend the study with experimental results will be followed by some additional computational work which is ongoing for designing more compounds with different mutations in DHFR, along with the compounds of our previous computational work published as  "In Silico Study Probes Potential Inhibitors of Human Dihydrofolate Reductase for Cancer Therapeutics." Journal of Clinical Medicine (2019.02). So that we will have sufficient data to be tested experimentally before to collaborate with other lab. We hope that respected reviewer will consider our response generously.